# Structural insights into binding-site access and ligand recognition by human ABCB1

Devanshu Kurre[1], Phuoc X Dang [ID][1,3], Le T M Le [ID][1,4], Varun V Gadkari[2] & Amer Alam [ID][1✉]

## Abstract

**ABCB1 is a broad-spectrum efflux pump central to cellular drug handling and multidrug resistance in humans. However, how it is able to recognize and transport a wide range of diverse substrates remains poorly understood. Here we present cryo-EM structures of lipid-embedded human ABCB1 in conformationally distinct apo-, substrate-bound, inhibitor-bound, and nucleotide-trapped states at 3.4–3.9 Å resolution, in the absence of stabilizing antibodies or mutations. The substrate-binding site is located within one half of the molecule and, in the apo state, is obstructed by the transmembrane helix (TM) 4. Substrate and inhibitor binding are distinguished by major TM rearrangements and their ligand binding chemistry, with TM4 playing a central role in all conformational transitions. Furthermore, our data identify secondary structure-breaking residues that impart localized TM flexibility and asymmetry between the two transmembrane domains. The resulting structural changes and lipid interactions that are induced by substrate and inhibitor binding can predict substrate-binding profiles and may direct ABCB1 inhibitor design.**

**Keywords** ABC Transporter; Multidrug Resistance; ABCB1/MDR1/p-glycoprotein; cryo-EM; Structural Biology
**Subject Categories** Pharmacology & Drug Discovery; Structural Biology

## Introduction

The ATP-binding cassette (ABC) transporter ABCB1, also known as Multidrug resistance protein (MDR)1 or P-glycoprotein (p-gp) is a ubiquitously expressed drug exporter that plays a key role in cellular drug handling (Borst and Elferink, 2002; Borst and Schinkel, 2013; Darwich et al, 2010; Fromm, 2004; Hodges et al, 2011; Leslie et al, 2005; Thiebaut et al, 1987; Ueda et al, 1987). Its pharmacological relevance makes it a key transporter in the Food and Drug Administration's guidance for all developmental drugs to be screened against (Fiedorczuk and Chen, 2022). ABCB1 activity can be a limiting factor in cancer chemotherapy (Bauer et al, 2010; Leonard et al, 2003; Ling, 1997; Robey et al, 2018; Ueda et al, 1987) and treatment of neurological disorders (Fromm, 2004; Loscher and Potschka, 2002; Schinkel et al, 1996; Sita et al, 2017; Storck et al, 2022; Xie et al, 1999) and has been increasingly implicated in accumulation of amyloid-beta peptides, a hallmark feature of Alzheimer's Disease(Storck et al, 2022). Despite its relevance, ABCB1's promise as druggable clinical target remains unrealized largely due to systemic toxicities and off target effects resulting from its inhibition (Robey et al, 2018; Tamaki et al, 2011). Understanding the detailed mechanisms by which ABCB1 recognizes and transports a wide range of structurally and chemically diverse substrates remains a major focus in biomedicine. Visualizing the underlying chemistry involved is key to designing more specific ABCB1 inhibitors and circumventing ABCB1 mediated efflux of a wide range of developmental drugs. However, despite long-term efforts, ABCB1 has so far remained notoriously averse to direct structural analysis without the use of antibody fragments and stabilizing mutations to aid conformational trapping.

ABCB1 is a type II ABC exporter/type IV ABC transporter with each transmembrane domain (TMD) comprising 6 transmembrane helices (TMs) and followed by a cytosolic nucleotide-binding domain (NBD). It is topologically arranged as a pseudo-symmetric domain-swapped dimer with the 4th and 5th TMs of each TMD making extensive contacts with the opposing TMDs and NBDs as first revealed by the structure of its bacterial homolog Sav1866 (Dawson and Locher, 2006). To date, the only structures of human ABCB1 determined are those of its hydrolysis-deficient mutant in the ATP bound outward facing (OF) state and those in complex with antigen-binding fragments (Fabs) from the inhibitory antibodies UIC2 (Alam et al, 2019) and MRK16 (Nosol et al, 2020). Key mechanistic questions about polyspecific substrate recognition and the drug transport cycle of ABCB1 therefore remain open. First, the nature of its Inward-Facing (IF) apo state remains unknown, leaving open the question of how substrates gain access to their respective binding site(s). Second, the binding chemistry governing differential substrate and inhibitor interactions with ABCB1 in the absence of conformational trapping by inhibitory Fabs remains unknown. Third, it is unclear what role sequence and structural asymmetry plays in ABCB1 function. Finally, while lipids have been implicated in modulation of ABCB1 structure and its interaction with ligands (Clay et al, 2015; Hegedus et al, 2015; Loo and Clarke, 2016; Szewczyk et al, 2015), the extent and specifics of these interactions remains largely unexplored.

To address the above-mentioned gaps in knowledge, we determined multiple structures of wild-type human ABCB1 in a lipid environment by single particle cryo-EM. Four distinct conformations

[1]The Hormel Institute, University of Minnesota, Austin, MN 55912, USA. [2]Department of Chemistry, University of Minnesota, Minneapolis, MN 55455, USA. [3]Present address: Department of Pharmacy—Inpatient, Mayo Clinic, Rochester, MN 55901, USA. [4]Present address: Department of Biochemistry and Molecular Biology, Mayo Clinic, Rochester, MN 55901, USA. ✉E-mail: aalam@umn.edu

of the transporter were observed including, for the first time, its IF apo and substrate-bound states. These structures allow us to map out the conformational transitions associated with ligand and nucleotide binding and visualize key differences in how substrates and inhibitors interact with the TMD. They highlight the concerted TM and NBD movements underlying ATP-coupled drug transport and regulation of binding site access and the complex interplay between lipid interactions and TM secondary structure breaks that impart tremendous TMD flexibility and overall conformational heterogeneity to human ABCB1 that has made its high-resolution structure determination difficult. Overall, our results offer fundamental insights into the mechanistic details of the ABCB1 drug transport cycle and its inhibition that will have significant implications for ABCB1 targeted therapeutic design in various medical applications as well as broader drug-development efforts where potential ABCB1 interactions may limit drug-bioavailability, among other undesired effects.

## Results

### Four distinct conformations of lipid-embedded wild-type human ABCB1

Human ABCB1 was stably expressed in HEK293 cells, purified in detergent, and reconstituted in saposin A (sapA) nanoparticles comprising a mixture of brain polar lipids (BPL) and cholesterol (Chol). SapA reconstituted ABCB1 displayed a more homogenous mass distribution as analyzed by native-mass spectrometry (nMS) as well as greater ATPase activity compared to MSP1D1 nanodisc reconstituted samples (Fig. 1A,B) and was chosen for cryo-EM analysis. We analyzed ABCB1 in its apo state and in the presence of ATP/Mg$^{2+}$ and either the substrate Taxol, representing turnover conditions similar to a recent analysis for human ABCG2 (Yu et al, 2021), or its third-generation inhibitor Zosuquidar. Taxol and zosuquidar complexes of ABCB1 in the absence of ATP/Mg$^{2+}$ displayed near identical conformations and are not discussed in further detail here. We also determined the structure of its nucleotide-trapped state in the presence of ATPγS, allowing for a visualization of the conformational spectrum associated with the drug transport cycle and its inhibition in ABCB1 (Fig. 1C). The overall conformation of the zosuquidar complex was nearly identical to the inhibitor occluded state seen in the presence of UIC2 or MRK16 Fabs (Alam et al, 2019; Nosol et al, 2020). Similarly, the ATPγS trapped ABCB1 structure was identical to that previously reported for the ATP-bound state of its hydrolysis-deficient EQ mutant in a detergent environment (Kim and Chen, 2018). In contrast, the conformations observed for its apo- and substrate-bound states are fundamentally different and have not been previously described. Conventional models of the apo state of ABCB1 based on homologous structures or alphafold predictions invoke a symmetric, IF conformation with a wide separation between the NBDs

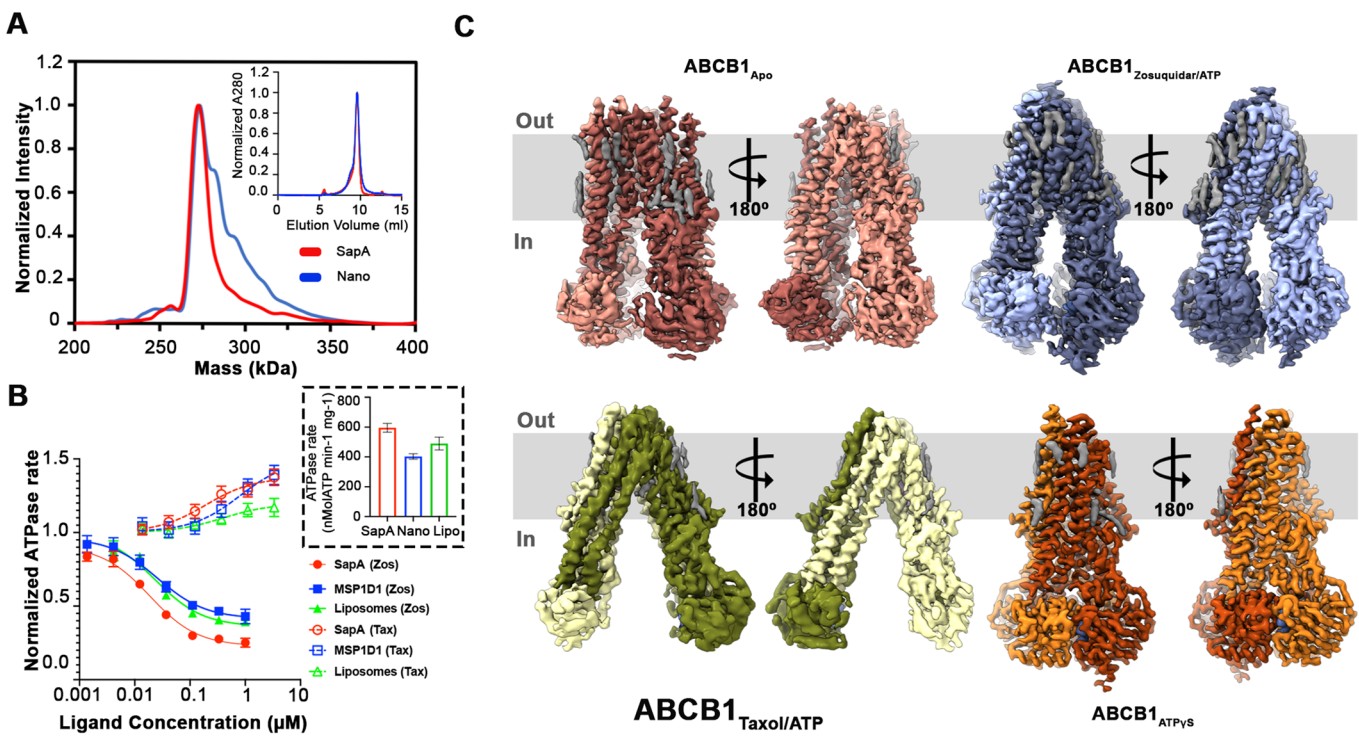

**Figure 1. Conformational landscape of lipid-embedded human ABCB1.**

(A) Comparison of saposin A and nanodisc reconstituted human ABCB1 by nMS. Normalized SEC chromatograms of both are shown in the top right corner. (B) Comparison of ATPase activity of saposin A, MSP1D1 nanodisc, and Liposome reconstituted human ABCB1 in the presence of inhibitor, Zosuquidar (solid shapes and lines) and Taxol (clear shapes and dashed lines), basal ATPase rates are shown in black dashed box. Data are presented as mean of experimental replicates ($N = 3$) and error bars denote standard deviation (center = mean). (C) Structures of human ABCB1 in multiple distinct conformational states. EM density for the two halves is colored differently with N-terminal half (half1) in lighter shade and C-terminal half (half2) in darker shade and that of modeled acyl chains is colored gray. Source data are available online for this figure.

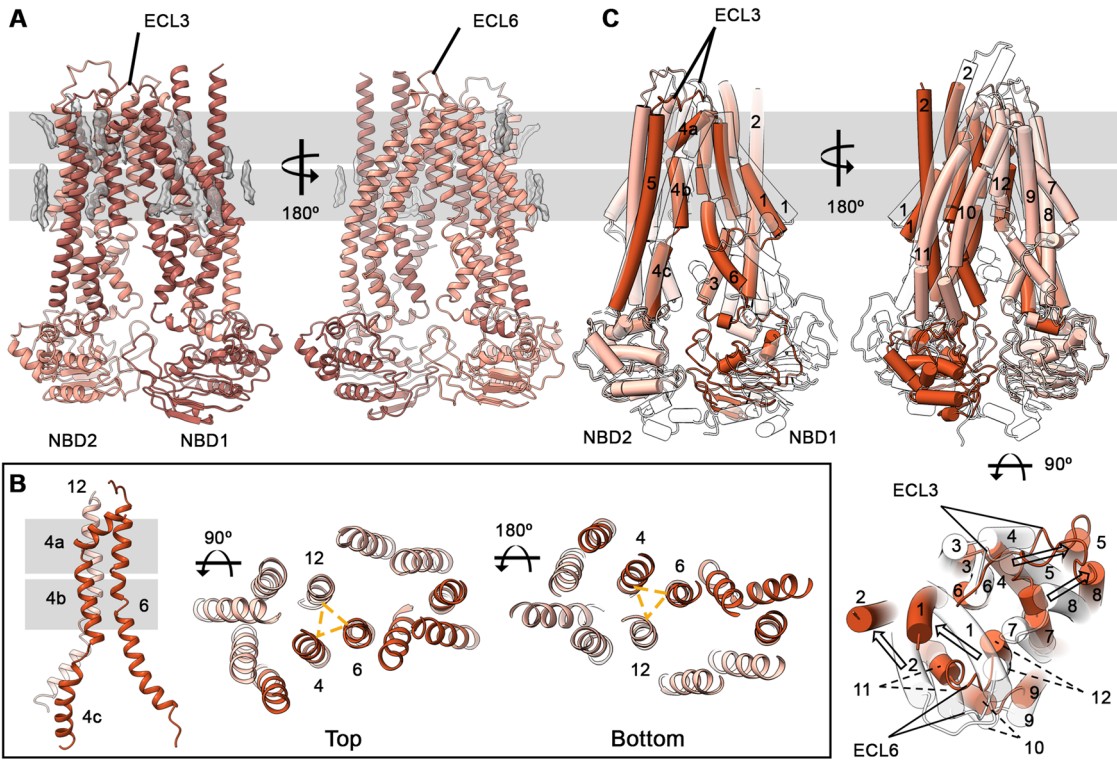

**Figure 2. Structure of apo-ABCB1.**

(A) Overall structure with the two halves colored as different shades of red and density modeled as lipid acyl chains (gray sticks) shown as transparent gray surfaces. (B) 3TM bundle formation by TM4, TM6, and TM12. TM4 sub-helical segments. The yellow dashed triangle highlights the central 3TM bundle in top and bottom views. (C) Comparison of the cryo-EM structure of apo-ABCB1, colored as in (A), and its alphafold-predicted structure (transparent cartoon). Black arrows indicate major movements of select TMs. The gray bars represent the plasma membrane.

as seen in the crystal structure of murine ABCB1 (Aller et al, 2009). Substrate binding is thought to promote NBD closure and explain consequent ATPase rate stimulation. In contrast, the apo state structure determined here displays distinct asymmetry between the two halves and closely spaced NBDs while the Taxol complex shows an IF$_{OPEN}$ state with wider NBD separation compared to the apo conformation, among other significant differences compared to structures of the Taxol complex of ABCB1 bound to inhibitory antibodies, as discussed in detail below.

## Apo ABCB1 adopts a unique IF$_{CLOSED}$ conformation

The predominant conformation of apo-ABCB1 observed here features an asymmetric TMD arrangement with a closed central TMD pathway (Fig. 2A), closely spaced NBDs, and widely spaced extracellular "wings" (Dawson and Locher, 2006) (Fig. 1C). We chose to classify this state as an IF$_{CLOSED}$ state based on TMD conformation. The structure is marked by multiple secondary structure (SS) breaks in the TMDs mediated by Glycine and Proline residues and several predicted SS breakers (Imai and Mitaku, 2005), most noticeably at G317 and G329 that leads to an elongation of extracellular loop (ECL)3 and wide separation between TM5 and TM6 (Fig. EV1). Conversely, ECL6, connecting TM11 and TM12 displays a lower degree of helix unraveling, likely owing to lower frequency of secondary structure-breaking residues that we speculate limit its conformational freedom and possibly that of

TM10 and TM11. As shown in Fig. 2B, closing of the central TMD pathway is facilitated by TM4, which adopts a kinked conformation with secondary structure breaks at P223 and K242, effectively dividing it into three sub-helices (TM4a-c). In conjunction with TM6 and TM12, it forms a central 3TM bundle that closes off the central cavity. In contrast to TM4, TM10 adopts a straight conformation, contributing further to structural asymmetry and leading to a lateral opening to the lower bilayer leaflet. These features lead to an overall conformation that diverges widely from canonical IF open conformations as demonstrated by a comparison to the alphafold-predicted structure of ABCB1 (Fig. 2C). The starkest differences are between the respective positions of TM1/ TM2 and TM4/TM5 pairs, leading to a more splayed open asymmetric arrangement of the extracellular "leaflets" and closer NBD spacing. The implications of this conformation for substrate and nucleotide access are expanded upon below.

## Distinct substrate and inhibitor interactions in human ABCB1

Previous analyses of substrate and inhibitor discrimination in human ABCB1 in the presence of conformational antibody Fabs revealed that both classes could occupy a centrally located, occluded TMD site with subtle differences between drug interacting residues and overall conformation (Alam et al, 2019; Nosol et al, 2020). Here we show that the predominant conformational states of

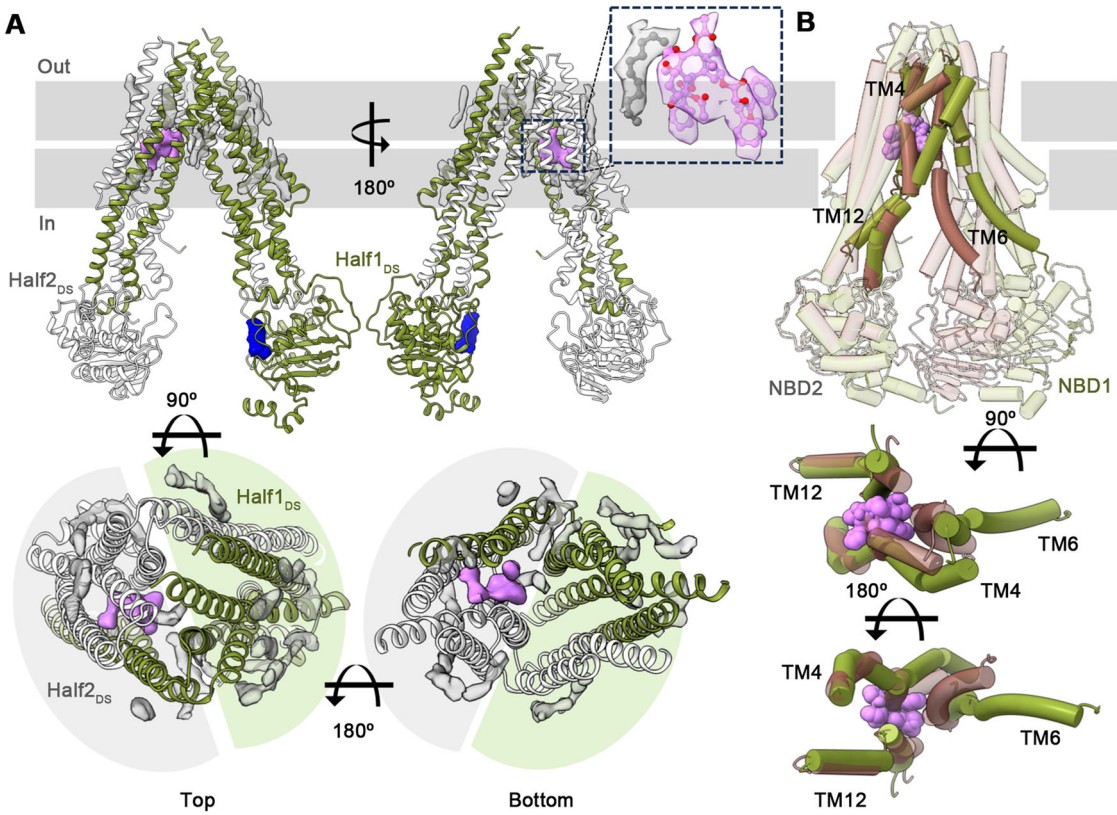

**Figure 3.  Structure of ABCB1 bound to Taxol.**

(A) Overall structure with first and second halves (primary structure based) colored green and white, respectively, and distinguished from domain-swapped (DS) halves. Density for Taxol and lipids is shown in pink and gray (0.01 contour threshold), respectively. The weaker density for the NBD1 nucleotide is shown in blue (0.008 contour threshold). The zoom panel shows Taxol (pink sticks) density along with associated density features modeled as a lipid acyl chain (gray sticks) as transparent pink and gray surfaces, respectively. Domain-swapped halves are highlighted and demarcated by gray and green semicircles. (B) Overall comparison of apo and Taxol complexes of ABCB1 (transparent brown and green cartoons, respectively) with 3TM forming helices (solid tube helices) and Taxol (pink spheres) is shown.

Taxol and zosuquidar complexes with ABCB1 alone are completely different. As shown in Fig. 3A, Taxol-bound ABCB1 adopts a symmetrical IF conformation with wider NBD spacing compared to the apo state. Taxol binding, however, is asymmetric, with a single molecule observed within the C-terminal half of the molecule/2nd half comprising the domain-swapped (DS) TMD2 (TM7-9 and TM12 from TMD2, and TM4 and TM5 from TMD1) and NBD2 pair, offset from the central TMD space. Interestingly, this binding site is occupied by TM4b in the apo state, which swings away to allow Taxol binding (Fig. 3B). This is accompanied by major rearrangements of TM5, ECL6, and TM6, breakup of the 3TM bundle observed in the apo state and an outward movement of NBD1. The position of NBD2 and its associated coupling helices remains largely unchanged. This links substrate binding to NBD orientation through TM4, which may act as an affinity gate to add a degree of substrate discrimination as expanded upon below. Density features within the hydrophobic TMD cavity are consistent with the presence of lipids and/or sterols. As their specific identity and orientation are impossible to ascertain from these data alone, we modeled them as unidentified acyl chains. A comparison of the two domain-swapped halves of Taxol-bound ABCB1 reveals distinct differences between residues within 5 Å of the observed Taxol molecule in the C-terminal half and its N-terminal equivalent

that would present a steric and electrostatic barrier to Taxol binding (Fig. EV2A,B).

In contrast to its Taxol complex, zosuquidar-bound ABCB1 adopts the same conformation as seen in the antibody Fab-bound complexes, marked by a fully occluded cavity with 2 closely interacting zosuquidar molecules (Fig. 4A,B). Cavity occlusion is brought about by the concerted kinking of TM4 and TM10, further highlighting its role in the overall transport cycle. Diffuse density for bound nucleotide is observed in NBD1. The overall structure of zosuquidar-bound ABCB1 shows increased positional order compared to the Taxol complex, with clearer density for TMD-associated lipids and NBD1-associated nucleotides. While the overwhelming majority of Taxol-interacting residues are drawn from the C-terminal half (Fig. 4C), zosuquidar interactions span both halves of the transporter (Fig. 4D), and no extraneous lipid density was observed in the occluded cavity.

## Structural transitions in human ABCB1 are asymmetric and dependent on TM4

The four conformational states of ABCB1 presented here allow for a direct comparison of the overall transitions associated with its drug transport cycle. As shown in Fig. EV3, the C-terminal half of

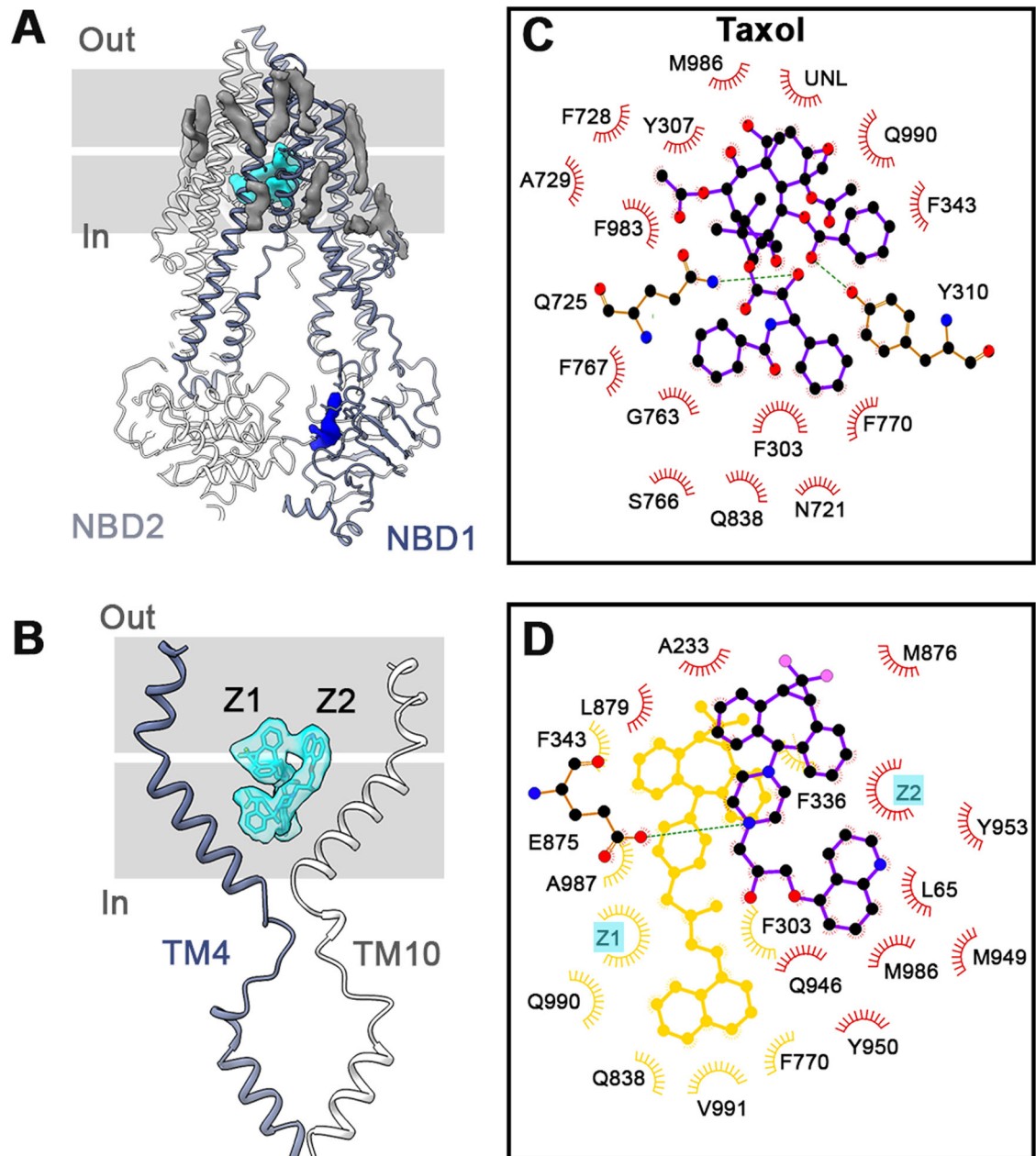

**Figure 4.  Comparison of Zosuquidar and Taxol binding.**

(A) Overall structure of ABCB1 bound to zosuquidar. Zosuquidar and ATP density is shown (0.0175 contour) as teal and blue surfaces, respectively. (B) Zoomed view of the occluded TMD cavity with TM4 and TM10 shown. EM density for both zosuquidar molecules (teal sticks, Z1 and Z2) is shown as a transparent teal surface (0.017 contour). (C) Ligand interaction plot of ABCB1 complexed to Taxol. (D) Ligand interaction plot of zosuquidar (z) bound ABCB1 with the second zosuquidar molecule is shown in yellow.

the transporter remains relatively rigid in comparison to its N-terminal counterpart, with significant positional changes of NBD1 associated with the different TMD conformations. Inter NBD separation is similar for the apo and inhibited state with the widest separation between the NBDs of the Taxol-bound IF conformation and narrowest separation for the sandwiched NBD dimer in the ATPγS complex as highlighted by Cα distance measurements between T263 (CH2) and R905 (CH4) (Fig. 5A, lower). While the overall conformations of the four states diverge

significantly, a pairwise alignment of TM pairs 1/2, 3/6, and 4/5 (and their half2 counterparts) shows expected patterns of linked movements during conformational cycling (Lee et al, 2014) with major exceptions for TM4 and TM10, and to a lesser extent, TM1 and TM2 (Fig. 5A,B). TM4 adopts a different conformation in all 4 structures, including three unique kinked conformations in the apo, substrate-bound, and inhibitor-bound states. Similarly, TM10 adopts different conformations in all four structures, but only the zosuquidar-bound state displays a kinked conformation like that of

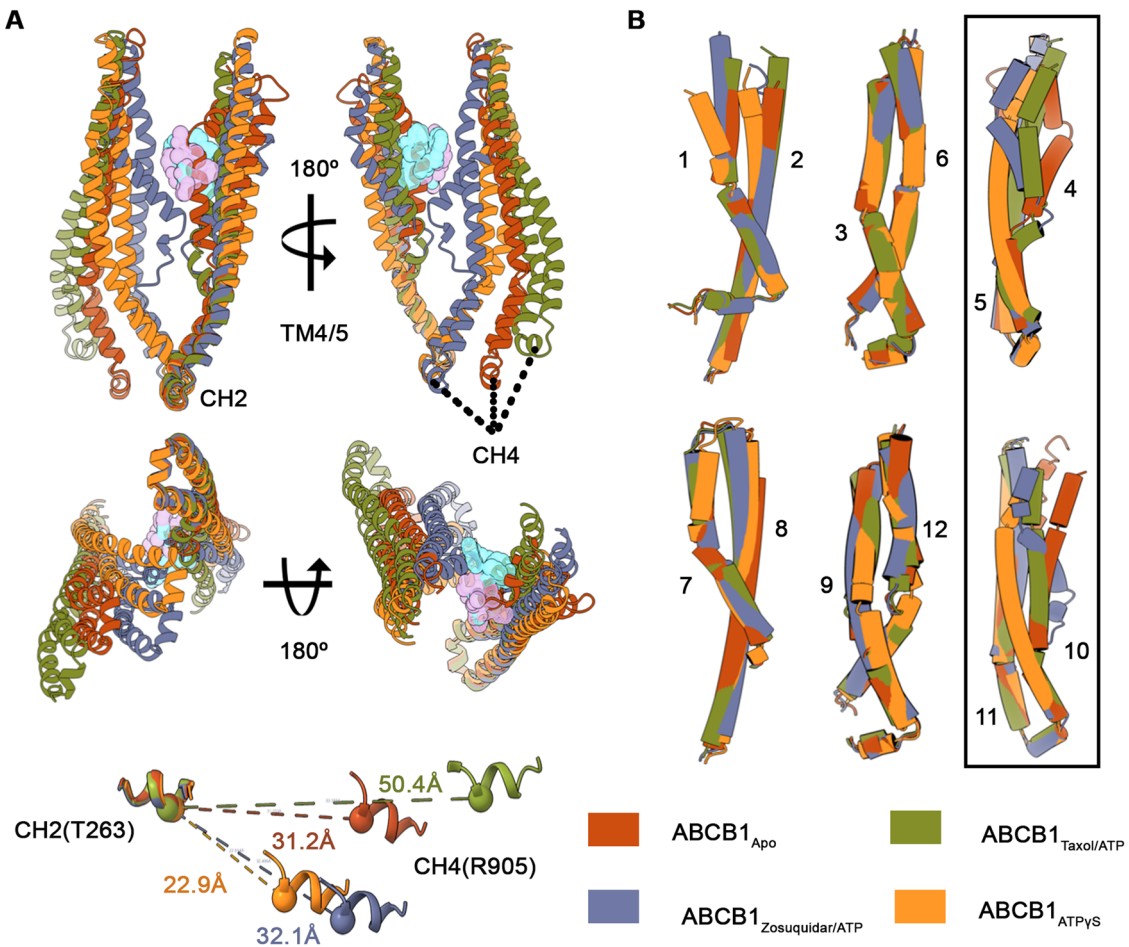

**Figure 5. Structural transitions in ABCB1.**

(A) Overlay of TM4/5 and TM10/11 of all ABCB1 structures reported, highlighting overall conformational changes linked to substrate (Taxol, pink surface) or inhibitor (zosuquidar, teal surface) binding and CH2 and CH4 movements (bottom) with distances between selected Cα pairs shown. (B) Pairwise structural alignment of linked TM pairs expected to move together in different type II ABC exporter conformational states with TM4/5 and TM10/11 pairs boxed to highlight their greater conformational flexibility in the four conformations reported.

TM4. The Cytoplasmic halves of all TMs match very closely in all structures, revealing that the conformational changes occur within the membrane environment, likely stabilized by dynamic lipid contacts, as expanded upon below.

## Discussion

Insights into TMD access and auto-inhibition of the binding site by TM4 gleaned from our data fundamentally change our understanding of how human ABCB1 works, allowing us to formulate an updated mechanism for substrate transport and its inhibition in ABCB1 as shown in Fig. 6. Central to this scheme is TM4, which acts as a gating helix and undergoes large-scale rearrangements in all conformations reported here. In the unbound (apo) state, human ABCB1 likely exists in a conformational equilibrium between multiple IF states. The $IF_{CLOSED}$ state that is dominant from our analysis is incompatible with substrate binding, with TM4 involved in 3TM bundle formation to close the TMD pathway and also sterically occluding the substrate-binding

site. As such, TM4 may play an autoinhibitory role and act as an affinity filter akin to the regulatory domains of ABCC-type transporters (Khandelwal and Tomasiak, 2024; Mao et al, 2024). Substrates overcoming this affinity threshold shift the conformational equilibrium towards an $IF_{OPEN}$ state with greater NBD separation, concurrent opening of the 3TM bundle, and ejection of TM4b from the substrate-binding site. Compared to the apo state, this NBD separation may be more sterically favorable for ATP binding (ATP/ADP exchange), linking substrate binding to stimulation of ATPase rates. Interestingly, the Taxol binding mode observed in the $IF_{OPEN}$ state overlaps with that for marine pollutants observed in the crystal structures of murine ABCB1 (Nicklisch et al, 2016), hinting at conserved patterns of ligand interactions. The $IF_{OCCLUDED}$ state observed in the zosuquidar complex reported here was previously shown to be stabilized by inhibitory antibody Fabs and capable of accommodating both inhibitors and substrates including Taxol (Alam et al, 2019; Alam et al, 2018; Nosol et al, 2020). The fact that Taxol alone could not be captured in the occluded state without stabilizing Fabs indicates that for substrates this conformation likely represents a sparsely

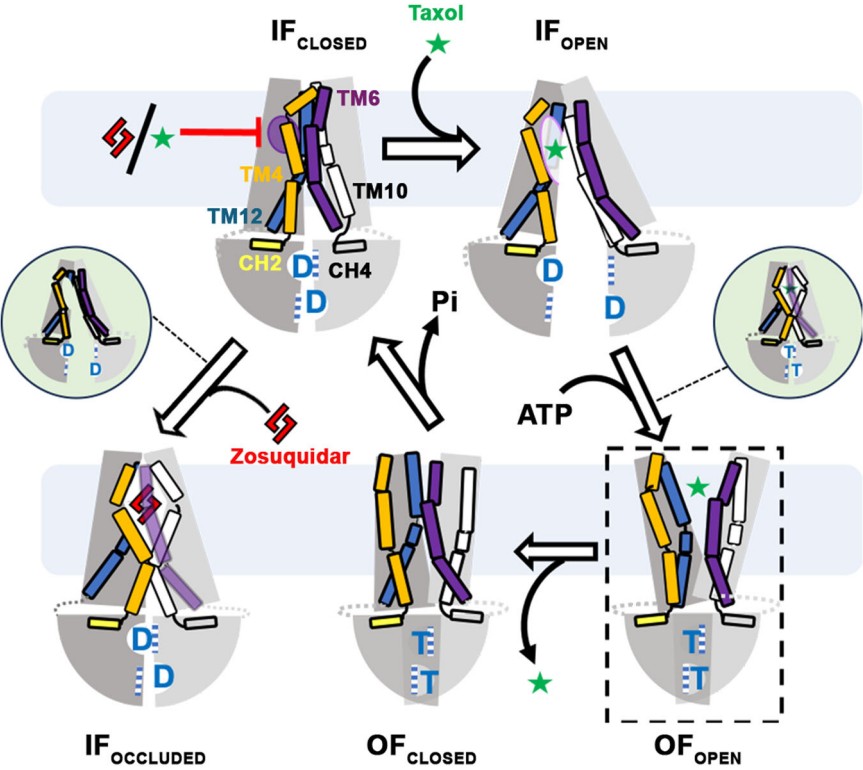

**Figure 6.  Mechanism of ABCB1 transport function.**

Schematic of working model for substrate transport and inhibition in human ABCB1: In the apo state an IF$_{CLOSED}$ state dominates with TM4 blocking the substrate-binding site. Substrate (Taxol, green star) binding promotes transition to the IF$_{OPEN}$ state, favoring ATP binding that leads to a transition to the OF$_{OPEN}$ state for substrate release through a preceding IF$_{OCCLUDED}$ state. Unlike substrates, inhibitors like zosuquidar (red L-shape) stabilize the IF$_{OCCLUDED}$ state, effectively trapping the transporter and arresting the transport cycle. Substrate release leads to the formation of an OF$_{CLOSED}$ state and ATP hydrolysis resets the transport cycle. With the exception of the OF$_{OPEN}$ state (based on homologous transporters like human ABCD1(16) and Sav1866(17)), all other states are based on experimentally determined structures. Select TMs driving conformational transitions are highlighted. Green circles highlight potential intermediate/alternate states. D = ADP, T = ATP. Pi inorganic phosphate. Dashed blue lines = ATP-binding elements.

populated, high-energy state prior to substrate extrusion through the OF$_{OPEN}$ conformation. Conversely, inhibitors like zosuquidar stabilize the IF$_{OCCLUDED}$ conformation, thereby inhibiting the transport cycle. This clear difference between substrates and inhibitors can be explained by their divergent ligand interactions. Asymmetry may play a key role here, with inhibitors like zosuquidar able to make stabilizing interactions with both domain-swapped halves of the transporter. In contrast, substrates like Taxol seen to bind within TMD2$_{DS}$ may be destabilized upon contact with TMD1$_{DS}$ upon ATP binding induced NBD closure and consequent TMD rearrangements, promoting a transition to the OF$_{OPEN}$ conformation and substrate extrusion. This suggests that TMD1$_{DS}$ residues that have been implicated in substrate interactions through mutagenesis and cellular efflux studies but not seen to directly interact with substrate here may be involved in promoting extrusion rather than stabilizing substrate binding (Chufan et al, 2016; Chufan et al, 2013; Loo and Clarke, 1998, 2008, 2015; Nasim et al, 2020). Upon substrate extrusion, the external leaflets of ABCB1 adopt a closer arrangement in contrast to OF states such as that seen in human ABCD1 (Le et al, 2022). Interestingly, this OF$_{CLOSED}$ state is also characterized by the formation of a 3TM bundle like that in the apo state, albeit involving TM6, TM7, and TM12, and may similarly serve to

prevent undesired substrate or lipid interactions before the transporter resets upon ATP hydrolysis to its IF conformation(s).

Our structural data reconcile decades of mutagenesis studies implicating residues involved in substrate interactions with distinct conformational states of human ABCB1. Notably, residues like F303, N721, F728, A729, S766, and Q838 that were not seen to closely interact with Taxol in the antibody Fab-bound occluded state were shown to do so in the Fab-free IO structure reported here (Becker et al, 2009; Chufan et al, 2013; Gao et al, 2020; Loo et al, 2006; Murakami et al, 2023; Nosol et al, 2020; Singh et al, 2014; Vahedi et al, 2017). A smaller subset of residues previously implicated in substrate interactions like I340, S344, and Q347 (Iqbal et al, 2022; Loo and Clarke, 2002; Sasitharan et al, 2021) were shown to interact with Taxol in the Fab-bound IF$_{OCCLUDED}$ state but not in the IO state reported here. Overall, the majority of Taxol-interacting residues are shared between the two conformations, showcasing the binding site plasticity that is a hallmark feature of human ABCB1. The data also have a number of important implications for the development of better ABCB1 inhibitors as well as drugs that bypass its substrate transport cycle. First, the IF$_{CLOSED}$ apo state lays the foundation for the development of a new class of ABCB1 inhibitors that could potentially trap it, thereby preventing substrate access to the TMD.

Second, the Taxol complex offers unprecedented detail into a discreet substrate-binding site that can aid the design of more accurate computational models for studying ABCB1 drug interactions. Third, the zosuquidar and Taxol complexes of ABCB1 define the underlying binding chemistry that distinguishes substrates and inhibitors. Finally, the data showcase the remarkable structural and functional variability helix-breaking elements impart to TMDs, especially in the context of a lipid bilayer environment and dynamic lipid interactions, that existing homology and predicted models have failed to capture. Additional structures of human ABCB1 in complex with drugs with different physiochemical properties are needed to explore the extent of binding site plasticity and potential deviations from the mechanistic framework proposed above.

# Methods

### Reagents and tools table

| Reagent/resource | Reference or source | Identifier or catalog number |
| --- | --- | --- |
| **Experimental models** | | |
| Flp-In T-REx 293 cell line | Thermo Fisher Scientific (TFS) | R78007 |
| **Recombinant DNA** | | |
| ABCB1 | TFS Geneart sequence, gift from Kaspar Locher Lab | |
| MSP1D1 | Addgene | |
| Saposin A | Salipro | |
| GFPnb | Addgene | |
| 3C protease | Addgene | |
| pcDNA 5/FRT | Thermo Fisher | V652020 |
| pOG44 | Thermo Fisher | V600520 |
| 1D4 peptide | GenScript | |
| **Antibodies** | | |
| Rho1D4 antibody | University of British Columbia | |
| **Oligonucleotides and other sequence-based reagents** | | |
| **Chemicals, enzymes, and other reagents** | | |
| DMEM | GIBCO | 10566-016 |
| 100 kDa centrifugal filters Amicon ultra-4 | MILLIPORE | UFC810096 |
| 1D4 peptide | GENSCRIPT | |
| Acetic acid | SIGMA-ALDRICH | A6283-1L |
| Ammonium acetate | SIGMA-ALDRICH | 431311 |
| Ammonium molybdate tetrahydrate | EMD CHEMICALS | AX1310-3 |
| Antimycotic antibiotic | GIBCO | 15240-062 |
| L(+) Ascorbic Acid | MERCK | AX1775-3 |
| ATP | SIGMA-ALDRICH | A7699 |
| Bio-Beads SM-2 | BIO-RAD | 1528920 |
| Brain polar extract lipids | AVANTI POLAR LIPIDS | 141101C-500MG |
| Cell culture dishes 145/20 mm | GREINER | 639160 |
| Cholesterol | SIGMA-ALDRICH | C8667 |
| Cholesterol hemisuccinate | ANATRACE | CH210 |
| CNBr activated 4B Sepharose resins | CYTIVA | 17043001 |

| Reagent/resource | Reference or source | Identifier or catalog number |
| --- | --- | --- |
| Blue Coommasie stain | ABCAM | AB119211 |
| DDM ANAGRADE | ANATRACE | D310 |
| DMSO | SIGMA-ALDRICH | D2650 |
| Fetal bovine serum | GIBCO | A56707 |
| Glycerol | MP BIOMEDICALS | 800689 |
| HCl | SIGMA-ALDRICH | 30721-1L |
| HEPES sodium salt | SIGMA-ALDRICH | H3784 |
| LB | FISHER | BP9731-500 |
| Difco LB agar | BD AND COMPANY | 244520 |
| $MgCl_2$ | SIGMA-ALDRICH | M2670 |
| NaCl | SIGMA-ALDRICH | S9888 |
| Ni-NTA superflow resin | QIAGEN | 30430 |
| Penicillin–streptomycin antibiotic | GIBCO | 15140-122 |
| PBS | QUALITY BIOLOGICAL | 119-069-101 |
| Potassium phosphate dibasic | SIGMA | 60353 |
| Protease inhibitor Complete O mini tablets | ROCHE | 11836170001 |
| Rho1D4 antibody | | |
| SDS | CALBIOCHEM | 7910 |
| SDS-PAGE precast gels | BIO-RAD | 4561093 |
| Sodium (meta)arsenite | ALDRICH | S7400 |
| Sodium citrate tribasic dihydrate | SIGMA-ALDRICH | S4641 |
| Taxol | PHYTOLAB | 89806 |
| Triton X-100 | SIGMA | T8787 |
| Zosuquidar | TOCRIS | 5456 |
| **Software** | | |
| Cryo-EM data processing software | Relion | Version 4.0 |
| Graph plot | GraphPad Prism | Version 10.4.0 |
| Ligand schematic diagram | LigPlot | Version v.2.2.8 |
| Molecule visualization | ChimeraX | Version 1.7.1 |
| HPLC | Agilent Technologies | Agilent 1260 Infinity II |
| Native MS software | Proteinaceous Inc. | STORIBoard |
| **Other** | | |

### Cell culture, protein expression, and purification

The expression and purification of wild-type human ABCB1 were conducted largely as previously described (Alam et al, 2019; Alam et al, 2018; Nosol et al, 2020). First, an ABCB1 stable cell line with a C-terminal eYFP-Rho1D4 tag and a 3C/precision protease site between the protein and tag was generated using the Flp-In TREX system (Thermo Fisher Scientific) for tetracycline-inducible expression. These ABCB1 stable cells were grown in DMEM media supplemented with 10% fetal bovine serum (FBS), penicillin–streptomycin, and antimycotic antibiotics at 37 °C in a 5% $CO_2$ incubator until they reached over 70% confluency, which typically took about 72–96 h. Next, the media was replaced with DMEM supplemented with 2% FBS and 0.6 µg/ml tetracycline, and

the cells were allowed to express the protein for 72 h at 37 °C in a 5% $CO_2$ incubator. These cells were subsequently washed with PBS before being harvested by centrifugation at 3000 r.c.f. for 3 min at 4 °C, and flash-frozen in liquid nitrogen for storage at −80 °C.

All protein purification steps were carried out at 4 °C or on ice. Cell pellets were thawed and resuspended in eight volumes of lysis buffer per gram of pellet (25 mM HEPES pH 7.5, 150 mM NaCl, 20% glycerol, 0.5 mM PMSF, 2 µg/ml trypsin inhibitor, and one complete mini tablet per 50 ml). After dounce homogenizing, the cell lysate incubated with a 0.5%/0.1% mixture of n-dodecyl-β-D-maltopyranoside (DDM) and cholesteryl hemisuccinate (CHS) for 2 h, and then centrifuged at 48,000 r.c.f. for 30 min. The supernatant was applied to Cyanogen bromide-activated Sepharose 4b beads (Cytiva) coupled to Rho1D4 antibody (University of British Columbia) resin for binding over 3 h. The resin was washed four times with 10 column volumes (CV) of wash buffer (25 mM HEPES pH 7.5, 150 mM NaCl, 20% glycerol, and 0.02%/0.004% DDM/CHS) followed by protein elution by addition of wash buffer supplemented either with 0.25 mg ml$^{-1}$ 1D4 peptide (GenScript) or a 1:10 w:w ratio of 3C protease for on-column cleavage and incubated overnight at 4 °C on a roller for tag cleavage. 3C protease was removed by incubation with Ni-NTA beads.

## Lipid reconstitution of ABCB1

Expression and purification of MSP1D1 and Saposin A was carried out as described (Frauenfeld et al, 2016; Ritchie et al, 2009), except that the final purification and storage buffer contained 25 mM HEPES pH 7.5, 150 mM NaCl. Expression plasmids for MSP1D1 and Saposin A were obtained from Addgene and Salipro Biotech AB, respectively. Brain Polar Extract lipids (BPL, Avanti) and cholesterol (Chol, Sigma) were mixed at an 80:20 w:w ratio and dried using a rotary evaporator (Bucchi), resuspended in diethyl ether, dried again, and finally resuspended in HEPES buffer (25 mM HEPES pH 7.5, 150 mM NaCl). Nanodisc reconstitution followed our published protocols (Le et al, 2022; Le et al, 2023). Briefly, The BPL/Chol mixture was solubilized in storage buffer supplemented with a 0.2%/0.04% solution of DDM/CHS and homogenized using water bath sonication, with three cycles of 2 min on and 2 min off. 3C cleaved or ID4 peptide eluted ABCB1 was mixed with MSP1D1 and solubilized lipids a molar ratio of 1:10:350 for ABCB1:MSP1D1:BPL/Chol and the mixture diluted to reduce the final glycerol concentration to less than 4% (v:v). After 1-h incubation at 4 °C on a roller, detergent was removed by addition of 0.8 grams/ml reaction buffer of Bio-Beads SM-2 (Biorad) prewashed in storage buffer and incubated on a roller for 2 h at room temperature (RT). The supernatant was removed from the biobeads and concentrated using a 100 kDA molecular weight cutoff (M.W.C.O) centrifugal filter. Saposin A reconstituted ABCB1 was prepared similarly except that a 1:15:400 molar ratio of ABCB1:Saposin A:BPL/Chol was used. Protein concentration was measured by densitometry analysis of SDS-PAGE bands using detergent-purified ABCB1 of known concentrations as standards.

ABCB1 proteoliposomes were prepared as described (Geertsma et al, 2008) with minor modifications. Briefly, the BPL/Chol lipid mixture (80:20 wt:wt ratio) was first reconstituted in buffer comprising 150 mM NaCl and 25 mM HEPES pH 7.5 at a concentration of 20 mg ml$^{-1}$. Empty liposomes were prepared through extrusion using a 0.2 µm filter. Pre-formed liposomes and detergent-purified ABCB1 were supplemented with at 0.3% and 0.14% (v:v) of Triton X-100, respectively, mixed, and incubated at RT for 30 min. Detergent removal was done in five successive incubation steps using each using fresh 50 mg Bio-beads SM-2 per ml reaction mix. The incubation steps were carried out with gentle agitation for 30 min at RT, 60 min at 4 °C, overnight at 4 °C, followed by two 60-min incubations at 4 °C. Liposomes were pelleted by ultracentrifugation at 80,000 r.p.m. using a TLA-100 rotor (Beckmann Coulter), the supernatant discarded and resuspended in an equivalent volume of reconstitution buffer at 0.5–1 mg ml$^{-1}$.

## ATPase assays

ATPase measurements were based on a molybdate-based calorimetric assay measuring release of inorganic phosphate (Pi) (Chifflet et al, 1988) as described (Le et al, 2022; Le et al, 2023). Stocks of zosuquidar (Tocris) and Taxol (PhytoLab) were prepared in 100% dimethylsulfoxide DMSO. ATPase measurements were performed by incubating 0.02–0.1 mg ml$^{-1}$ ABCB1 with 2 mM ATP, 10 mM $MgCl_2$ with varying concentrations of zosuquidar or Taxol at 37 °C. Statistical analyses and linear regression were done in GraphPad Prism 9. ATPase assays were done as experimental replicates (*n*) of 3 for the zosuquidar and Taxol and 12 for the absolute ATPase rate comparisons.

## Native-mass spectrometry

Wild-type ABCB1 was purified and reconstituted into nanoparticles as described in the above sections. The detergent sample and the reconstituted ABCB1 samples were buffer exchanged into 200 mM ammonium acetate (99.999% Trace Metals Basis, Sigma Aldrich) containing 0.02% DDM/0.004% CHS (only 200 mM ammonium acetate for nanoparticle sample) using 40k zeba spin desalting column and further purified by injecting into an Agilent 1260 Infinity II LC system (Agilent Technologies) using preequilibrated TSKgel G4000SWxl column (TOSOH Biosciences).

Samples were diluted to 500 nM and ionized via nanoelectrospray ionization using gold-coated borosilicate capillaries (prepared in-house) and analyzed on a Q Exactive Ultra High Mass Range orbitrap mass spectrometer (Thermo Fisher Scientific) (Fort et al, 2017; Wilm and Mann, 1996). The instrument was operated in Direct Mass mode, enabling orbitrap-based charge detection mass spectrometry measurements of individual intact lipoprotein nanoparticle ions (Kafader et al, 2019; Worner et al, 2020). Briefly, the instrument was operated with the Ion Target set to "high *m/z*" and the Detector Optimization set to "low *m/z*." The in-source trapping and higher-energy collisional dissociation cell were operated at 1–10 V. All measurements were acquired at a resolution setting of 200,000 (FWHM at *m/z* 400) with a trapping gas pressure setting of 1. All data processing was performed using STORIBoard (Proteinaceous Inc.). Ions were filtered based on ion lifetime and signal-to-noise, and ion charge states were assigned using the "Voting v3" charge assignment algorithm (Kafader et al, 2019). Ion filtering and charge assignment parameters are summarized in Table EV1. Charge assignment was calibrated using carbonic anhydrase, alcohol dehydrogenase, pyruvate kinase, beta-galactosidase, and GroEl. All samples were acquired for 10–20 min, and the reported measurements are representative of ~10,000 ions.

## Cryo-EM sample preparation and data collection

For Grid preparation ABCB1-eYFP reconstituted in Saposin A Nanoparticles (SapNPs) were incubated antiGFP nanobody (Addgene) coupled Sepharose 4B resin prepared in-house for 2 h at 4 °C, washed with 3 ×10CV of reconstitution buffer, followed on-column cleavage by in 3CV reconstitution buffer supplemented with 3C protease to recover ABCB1 SapNPs. Samples were subsequently concentrated using a 100 MWCO centrifugal filter and further purified by Size exclusion chromatography (SEC) on an Agilent 1260 Infinity II LC system (Agilent Technologies) using a TSKgel G4000SWxl column (TOSOH biosciences) pre-equilibrated with reconstitution buffer at 4 °C. Pooled peak fractions from SEC at a concentration of ~0.15 mg ml$^{-1}$ (~1 μM) were mixed with a 10× molar excess of Taxol or zosuquidar with or without ATP/Mg$^{2+}$ and incubated together before concentrating the samples to 0.5–1 mg ml$^{-1}$ (~3–6 μM) for grid preparation similar to prior studies(Alam et al, 2019). 4 μL of sample was applied to the glow discharged (60 s, 15 mA) Quantifoil R1.2/1.3 Cu grids using Vitrobot Mark IV with 4 s blot time and 0 blot force under >90% humidity at 4 °C and plunge frozen in liquid ethane. All grids were clipped and stored in liquid nitrogen.

All the cryo-EM data were collected on a 300 kV Titan Krios electron microscope equipped with a Biocontinuum K3 Direct Electron Detector with 20 eV GIF energy filter, 50 eV condenser C2 and 100 μm objective apertures. Automated data collection was carried out using the EPU 2.8.0.1256REL software package (Thermo Fisher Scientific) at a magnification of 130,000× in Counted Super Resolution mode corresponding to a calibrated pixel size of 0.664 Å with defocus range set from −0.5 μm to −2.5 μm. Three shots were taken per hole. Image stacks comprising 40 frames were recorded for 60 s at an estimated dose rate of 1e-/Å$^2$/frame.

## Data processing, model building, and refinement

Data processing was done in Relion (Kimanius et al, 2024; Scheres, 2012; Zivanov et al, 2022). In brief, image stacks were motion-corrected using Relion's internal MotionCor2 implementation, followed by CTF estimation using CTFFIND4 (Rohou and Grigorieff, 2015). All resolution estimates were based on the gold standard 0.143 cutoff criterion (Scheres, 2012). Data collection and processing parameters are provided in Table EV2 along with model building and refinement statistics. Data processing flow charts are shown in Appendix Fig. S1. EM density around individual domains/TMs, FSC curves, and Local resolution-colored maps are shown in Appendix Figs. S2, S3, and S4, respectively.

For ABCB1-apo, an initial dataset comprising 5974 micrographs was used for reference-free automated particle picking (Laplacian-of-Gaussian algorithm) within Relion. In all, 2,167,202 particles were extracted at a 3× binned pixel size of 1.992 Å and subjected to several rounds of 2D classification, followed by Ab-initio model building using within Relion. This initial model was used for subsequent 3D classification (number of classes (N) = 5) and a single predominant class comprising 662,694 was refined to 5.1 Å followed by another round of 3D classification (N = 5) and 3D refinement, re-extraction at a 1.5× binned pixel size of 0.996 Å, and particle polishing to yield a 4.1 Å map. A second set of 6,321,903 particles from 13,327 micrographs was picked using Topaz (default

model) and processed similarly except that a refined 3D class from the first set was used as a reference. A refined 3D at 4.0 Å resolution and comprising 660,276 particles was obtained. Particles from the final refined classes from both sets were combined, followed by additional rounds of 3D refinement and postprocessing to yield a 3.8 Å map.

For the ABCB1$_{Taxol/ATP}$ complex, 15,494,460 particles from 33,055 micrographs were autopicked using Topaz and extracted at a 3× binned pixel size of 1.992 Å. After one round of 2D Classification, 6,254,156 particles were used for 3D classification (N = 5) with a low-pass filtered ABCB1-apo map as a reference. The single highest resolution class revealed an IF conformation and was subjected to iterative 3D refinement and particle polishing, followed by subtraction of the SapNP. After 3D classification (N = 5), 154,538 particles from the highest resolution were reverted to their original non-subtracted images and refined to 3.9 Å.

For the ABCB1$_{Taxol}$ complex, 5725 micrographs were used to pick 2,547,172 particles by Topaz and extracted at a 3× binned pixel size of 1.992 Å. After 2D classification, 1,270,596 particles were used for 3D classification (N = 3) using the ABCB1$_{apo}$ map as a reference. In all, 486,111 particles from the best class were subjected to another round of 3D classification. The single highest resolution class comprised 133,895 particles and was refined to 4.7 Å.

For the ABCB1$_{Zosuquidar}$ complex, 2,182,930 particles were automatically picked by Topaz from 7281 micrographs. After two rounds of 2D classification, 943,398 particles entered 3D classification (N = 5) with a low-pass filtered ABCB1$_{apo}$ map used as a reference The single, highest resolution class comprising 373,279 particles was subjected to re-extraction at a 1.5× binned pixel size of 0.996 Å and signal subtraction to remove delocalized bulk lipid density and refined to 3.6 Å resolution.

For the ABCB1$_{Zosuquidar/ATP}$ complex, 10,710,935 particles from 12,897 micrographs were picked using topaz. In total, 2,468,729 particles were chosen for 3D classification (N = 5) using the map of the zosuquidar complex without ATP as a reference. A single highest resolution class comprising 733,688 particles was subjected to iterative rounds of 3D classification and particle polishing within Relion to yield a final refined map at 3.6 Å resolution.

Fort the ABCB1$_{ATPγS}$ sample, 7,689,616 particles from 12,165 micrographs were automatically picked using Topaz. After several rounds of 2D classification, 1,732,065 particles were subjected to 3D classification (N = 5) using a low-pass filtered ABCAB1$_{Apo}$ map as a reference. A single OF classes comprising 400,787 particles was subjected to another round of 3D classification (N = 5). 180,163 Particles from two similar and roughly equally populated OF classes were combined, re-extracted at a 1.5× binned pixel size of 0.996 Å and refined to 3.75 Å. A second dataset of 6,204,620 particles from 9318 micrographs was processed similarly to yield a final refined class at 3.5 Å comprising 260,172 particles. Particles from the final class from both datasets were combined and subjected to another round of 3D classification (N = 5) and the highest resolution class comprising 136,896 particles was refined to 3.4 Å.

Final EM maps were used for model building in COOT 0.9.6 EL (Brown et al, 2015). De novo model building was guided by the predicted structure of ABCB1 from AlphaFold2 (Jumper et al, 2021) for the apo and Taxol complexes. For the zosuquidar complexes, model building was guided initially by the structure of ABCB1 bound to the MRK16 fab (PDBID 7A6F). For the ATPγS complexed ABCB1,

the structure of ATP-bound ABCB1-EQ (PDBID: 6C0V) was used as an initial model before minor adjustments and refinement. Non-proteinaceous continuous density features attributed to lipids or sterols were modeled as Acyl chains. The structures were refined with secondary structure and geometry restrains in COOT 0.9.6 and PHENIX (Adams et al, 2010). Where NBD density was too weak for de novo model building, docked NBDs from higher resolution structures reported here were used and minimally refined. The final models for ABCB1$_{apo}$ comprised residues 33–81, 106–606, 694–1230, for ABCB1$_{Taxol/ATP}$ comprised residues 30–87, 100–630, 689–1257, for ABCB1$_{Zosuquidar/ATP}$ comprised residues 30–90, 104–630, 691–1272, and for ABCB1$_{ATP\gamma S}$ comprised residues 35–80, 105–630, 692–1276. Map and Structure visualization was performed in UCSF Chimera (Pettersen et al, 2004) and ChimeraX (Pettersen et al, 2021).

## Data availability

Requests for materials should be addressed to Amer Alam. The cryo-EM Maps have been deposited at the Electron Microscopy Databank (EMDB) under accession codes EMD-45854 (ABCB1$_{apo}$), EMD-45904 (ABCB1$_{Taxol/ATP}$), EMD-45903 (ABCB1$_{Zosuquidar/ATP}$), and EMD-45906 (ABCB1$_{ATP\gamma S}$) and the associated atomic coordinates have been deposited at the Protein Databank (PDB) under accession codes 9CR8, 9CTF, 9CTC, and 9CTG, respectively. Maps for ABCB1$_{Taxol}$ and ABCB1$_{Zosuquidar}$ have been deposited at the EMDB with accession codes EMD-45931 and EMD-45932, respectively.

The source data of this paper are collected in the following database record: biostudies:S-SCDT-10_1038-S44318-025-00361-z.

## Peer review information

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

## Acknowledgements

The authors would like to thank Dr. Kaspar Locher at ETH, Zurich, Switzerland, for providing the synthetic gene construct of human ABCB1. We would also like to thank the cryo-EM and shared instruments core facilities at the Hormel Institute for help with the experimental setup, and Dr. Jeppe Olsen, Dr. Jarrod French, Dr. Thanuja Sudasinghe, Dr. Subhrajyoti Dolai, and Ashley Wise for critical reading and discussion during manuscript preparation. This work was supported in part by the Hormel Foundation (Institutional research funds to AA), the National Institutes of Health (NIH) 1R01GM146906 (to AA), the Eagles Telethon postdoctoral fellowship (LTML and DK). VVG acknowledges funding from University of Minnesota start-up funds.

## Author contributions

**Devanshu Kurre**: Conceptualization; Data curation; Formal analysis; Validation; Investigation; Visualization; Methodology; Writing—original draft; Writing—review and editing. **Phuoc X Dang**: Data curation; Writing—review and editing. **Le T M Le**: Data curation; Writing—review and editing. **Varun V Gadkari**: Data curation; Formal analysis; Funding acquisition; Investigation; Methodology; Writing—original draft; Writing—review and editing. **Amer Alam**: Conceptualization; Data curation; Formal analysis; Supervision; Funding acquisition; Validation; Investigation; Visualization; Methodology; Writing—original draft; Project administration; Writing—review and editing.

Source data underlying figure panels in this paper may have individual authorship assigned. Where available, figure panel/source data authorship is listed in the following database record: biostudies:S-SCDT-10_1038-S44318-025-00361-z.

## Disclosure and competing interests statement

The authors declare no competing interests.

# Expanded View Figures

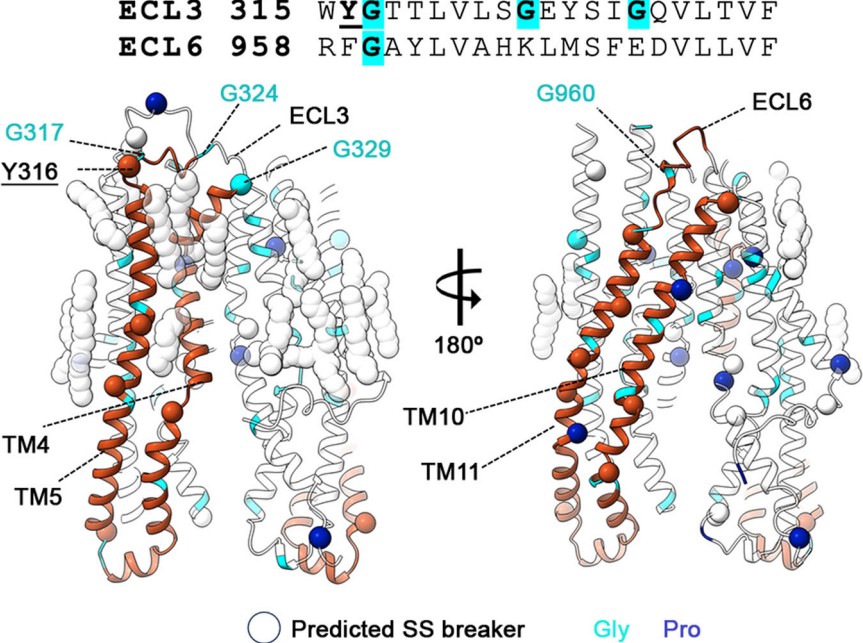

**Figure EV1. Secondary structure (SS) breaks in apo-ABCB1.**

Gly and Pro residues colored teal and blue, respectively, and predicted SS breaks shown as spheres. An ECL3 and ECL6 sequence alignment is also shown with residues colored similarly and predicted SS-breaking residues underlined. TM4/5 and TM10/11 pairs are colored red. Acyl chains for prospective lipid/sterol molecules are shown as transparent spheres.

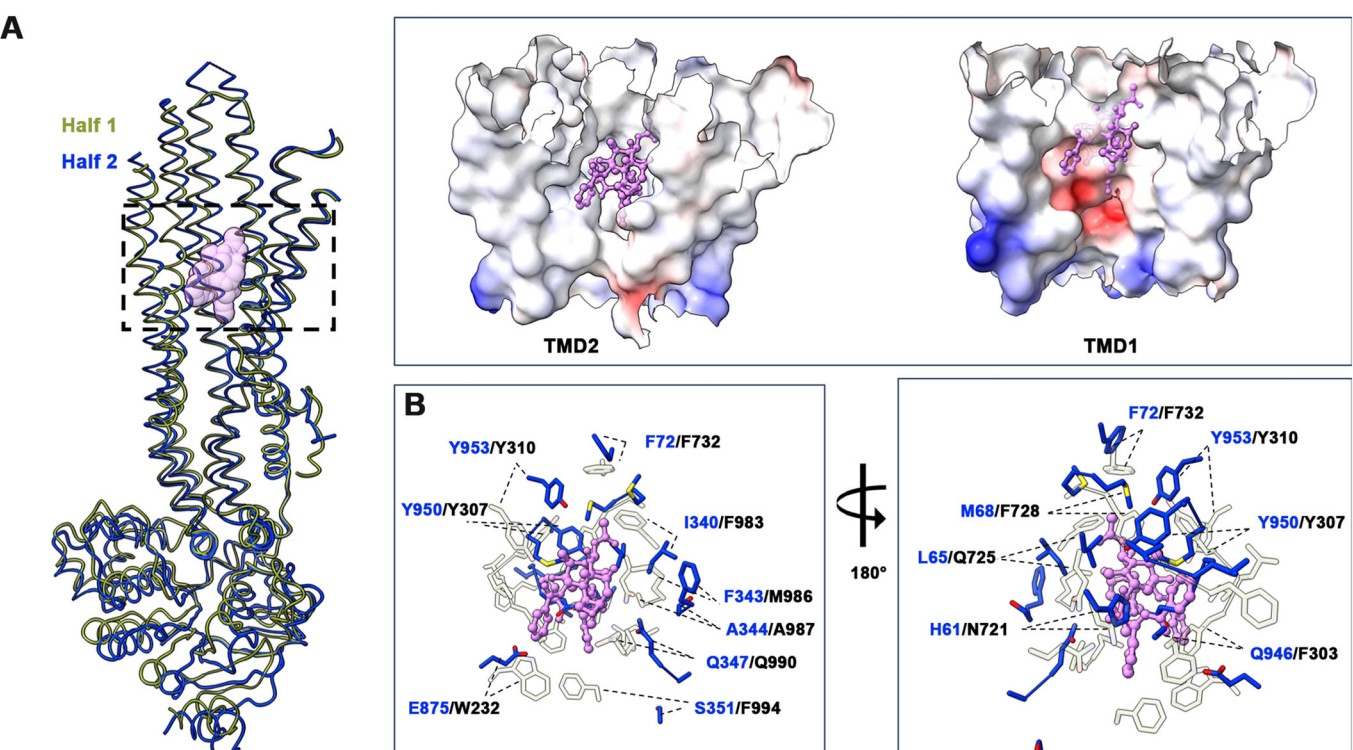

**Figure EV2.  Mismatch between TMD1 and TMD2 cavities for Taxol binding.**

(A) Overlay of domain-swapped (DS) halves of ABCB1. The Taxol molecule bound to TMD2$_{DS}$ is shown as transparent pink spheres. The Zoom panel shows electrostatic potential map of the TMD2$_{DS}$ cavity (left) and its TMD1$_{DS}$ cavity equivalent (right) showing electrostatic and steric clashes with Taxol. (B) TMD1$_{DS}$ equivalent residues of TMD2$_{DS}$ residues (Blue sticks) within 5 Angstroms of bound Taxol (transparent sticks), with residue labels colored similarly.

|  | ABCB1$_{\text{Taxol/ATP}}$ | ABCB1$_{\text{Zosuquidar/ATP}}$ | ABCB1$_{\text{ATP}\gamma\text{S}}$ |
|---|---|---|---|
| **ABCB1$_{\text{Apo}}$** | 1087 residue pairs: 13.593Å<br>374pruned atom pairs: 1.065Å | 1087 residue pairs: 10.186Å<br>357pruned atom pairs: 1.082Å | 1085 residue pairs: 11.585Å<br>171pruned atom pairs: 1.109Å |
| **ABCB1$_{\text{Taxol/ATP}}$** |  | 1155 residue pairs: 14.008Å<br>381pruned atom pairs: 1.098Å | 1138 residue pairs: 19.020Å<br>317pruned atom pairs: 1.168Å |
| **ABCB1$_{\text{Zosuquidar/ATP}}$** |  |  | 1153 residue pairs: 8.081Å<br>137pruned atom pairs: 1.314Å |

**Figure EV3. Overlay of different conformational states of ABCB1.**

Overall structural alignments between each conformation. R.m.s.d. values are also shown for total and aligned C alpha pairs.

