## [Peer Review File · The EMBO Journal]

Structural insights into binding-site access and ligand recognition by human ABCB1

Devanshu Kurre, Phuoc Dang, Le Le, Varun Gadkari, and Amer Alam

Corresponding author(s): Amer Alam (aalam@umn.edu)

Review Timeline:

Submission Date:	12th Aug 24
Editorial Decision:	4th Sep 24
Revision Received:	25th Nov 24
Editorial Decision:	3rd Dec 24
Revision Received:	9th Dec 24
Accepted:	13th Dec 24

Editor: William Teale

Transaction Report:

Dear Amer,

Thank you again for the submission of your manuscript entitled "Structural insight into binding site access and ligand recognition by human ABCB1" and for your patience during the review process. Your work was sent to two referees; we have now received the reports from both of them, which I copy below.

As you can see from their comments, both referees found your work to be of high technical quality and predict that it will be of use and interest to a wide field of research.

Based on the overall interest expressed in the reports, I would like to invite you to address the comments of all referees in a revised version of the manuscript. This revised version must take into account the important caveat of referee 1: namely that the physiological relevance of the key amino acids in the novel inhibitor-bound structure presented must be convincingly tested.

The EMBO Journal generally allows only a single major round of revision and that it is therefore important to be able to commit to such additional experiments at this stage. I believe this concern is reasonable and addressable, but please contact me if you have any questions, need further input on the referee comments or if you anticipate any problems in addressing any of their points. Discussing all of this by Zoom call is an open offer. Please, follow the instructions below when preparing your manuscript for resubmission.

I would also like to point out that as a matter of policy, competing manuscripts published during this period will not be taken into consideration in our assessment of the novelty presented by your study ("scooping" protection). We have extended this 'scooping protection policy' beyond the usual 3 month revision timeline to cover the period required for a full revision to address the essential experimental issues. Please contact me if you see a paper with related content published elsewhere to discuss the appropriate course of action.

Again, please contact me at any time during revision if you need any help or have further questions.

Thank you very much again for the opportunity to consider your work for publication. I look forward to your revision.

Best regards,

William

William Teale, Ph.D.
Editor
The EMBO Journal

When submitting your revised manuscript, please carefully review the instructions below and include the following items:

- 1) a .docx formatted version of the manuscript text (including legends for main figures, EV figures and tables). Please make sure that the changes are highlighted to be clearly visible.
- 2) individual production quality figure files as .eps, .tif, .jpg (one file per figure).
- 3) a .docx formatted letter INCLUDING the reviewers' reports and your detailed point-by-point response to their comments. As part of the EMBO Press transparent editorial process, the point-by-point response is part of the Review Process File (RPF), which will be published alongside your paper.
- 4) a complete author checklist, which you can download from our author guidelines ([https://wol-prod-cdn.literatumonline.com/pb-assets/embo-site/Author Checklist%20-%20EMBO%20J-1561436015657.xlsx](https://wol-prod-cdn.literatumonline.com/pb-assets/embo-site/Author%20Checklist%20-%20EMBO%20J-1561436015657.xlsx)). Please insert information in the checklist that is also reflected in the manuscript. The completed author checklist will also be part of the RPF.
- 5) Please note that all corresponding authors are required to supply an ORCID ID for their name upon submission of a revised manuscript.
- 6) We require a 'Data Availability' section after the Materials and Methods. Before submitting your revision, primary datasets

produced in this study need to be deposited in an appropriate public database, and the accession numbers and database listed under 'Data Availability'. Please remember to provide a reviewer password if the datasets are not yet public (see <https://www.embopress.org/page/journal/14602075/authorguide#datadeposition>). If no data deposition in external databases is needed for this paper, please then state in this section: This study includes no data deposited in external repositories. Note that the Data Availability Section is restricted to new primary data that are part of this study.

Note - All links should resolve to a page where the data can be accessed.

8) For data quantification: please specify the name of the statistical test used to generate error bars and P values, the number (n) of independent experiments (specify technical or biological replicates) underlying each data point and the test used to calculate p-values in each figure legend. The figure legends should contain a basic description of n, P and the test applied. Graphs must include a description of the bars and the error bars (s.d., s.e.m.).

9) We would also encourage you to include the source data for figure panels that show essential data. Numerical data can be provided as individual .xls or .csv files (including a tab describing the data). For 'blots' or microscopy, uncropped images should be submitted (using a zip archive or a single pdf per main figure if multiple images need to be supplied for one panel). Additional information on source data and instruction on how to label the files are available at .

10) We replaced Supplementary Information with Expanded View (EV) Figures and Tables that are collapsible/expandable online (see examples in <https://www.embopress.org/doi/10.15252/embj.201695874>). A maximum of 5 EV Figures can be typeset. EV Figures should be cited as 'Figure EV1, Figure EV2" etc. in the text and their respective legends should be included in the main text after the legends of regular figures.

12) Our journal encourages inclusion of *data citations in the reference list* to directly cite datasets that were re-used and obtained from public databases. Data citations in the article text are distinct from normal bibliographical citations and should directly link to the database records from which the data can be accessed. In the main text, data citations are formatted as follows: "Data ref: Smith et al, 2001" or "Data ref: NCBI Sequence Read Archive PRJNA342805, 2017". In the Reference list, data citations must be labeled with "[DATASET]". A data reference must provide the database name, accession number/identifiers and a resolvable link to the landing page from which the data can be accessed at the end of the reference. Further instructions are available at .

13) In order to increase the reproducibility and reach of your work, The EMBO Journal includes a table of reagents that were used in the study. Please provide this along with your revisions.

Further instructions for preparing your revised manuscript:

- a point-by-point response to the referees' comments, with a detailed description of the changes made (as a word file).
- a word file of the manuscript text.
- individual production quality figure files (one file per figure)
- a complete author checklist, which you can download from our author guidelines (<https://www.embopress.org/page/journal/14602075/authorguide>).

- Expanded View files (replacing Supplementary Information)

We realize that it is difficult to revise to a specific deadline. In the interest of protecting the conceptual advance provided by the work, we recommend a revision within 3 months (3rd Dec 2024). Please discuss the revision progress ahead of this time with the editor if you require more time to complete the revisions. Use the link below to submit your revision:

Referee #1:

In the submitted manuscript, Kurre et al. report different single particle cryo-EM structures of ABCB1. Human ABCB1, formerly called MDR1 or P-gp, is of prime importance as its overexpression lends cancer cells resistance to many chemotherapeutics posing a serious problem in the fight against cancer. Obviously, ABCB1 was the target of many studies including structure determination or the development of potent inhibitors. Despite excellent results in the laboratory, clinical success of these inhibitors was extremely limited.

Structures obtained by X-ray crystallography and single particle cryo-EM have provided important insights. Most importantly, work of the Locher group has demonstrated that substrates and inhibitors can be distinguished by their mode of binding. While substrates bind at a single site of ABCB1, inhibitors bind in pairs. This mode locks ABCB1 in a transport incompetent state. These structures were obtained for ABCB1 reconstituted in nanodiscs in the presence of a Fab fragment of an inhibitory antibody. A second studies with similar results were obtained for human-mouse chimeric ABCB1 again in the presence of an inhibitory antibody. Interesting to note that the senior author of this manuscript is an author of both of the above mentioned publications from the Locher lab.

Now, structures of ABC1 in the apo, substrate- and inhibitor-bound as well as nucleotide-trapped states at resolution between 3.4 to 3.9 Å are presented. ABCB1 was reconstituted this time in saposin A and structures were obtained in the absence of inhibitory antibodies. The structures of of the inhibitor-bound and nucleotide-trapped states are virtually identical to the published ones, while the apo and substrate-bound state showed profound differences. In the apso state the NBDs were closely spaced and a distinct asymmetry is apparent for both TMDs. The substrate-bound state displayed wider separated NBDS and a substrate that is bound asymmetrically to TMD2. And TMH4 plays a major role is preventing or providing access to the substrate binding side. This helix, which is split in three pieces acts as a gate keeper and as proposed by the authors selects substrates based on their affinity towards ABCB1.

From a cryo-EM point of view this is all fine. But in my opinion we are now confronted with a dilemma. We have two fundamentally different substrate-bound structures, one in the absence and one in the presence of an inhibitory antibody. Which one is right? Without additionally biochemical evidence, no one can answer this question.

Thus, the authors have to provide biochemical evidence (mutants, binding affinities, ATPase activity, etc.) that support their proposal before the manuscript can be accepted. Only then it s worthwhile to also mention minor points that deserve attention.

Referee #2:

The paper by Kurre et al., "Structural insight into binding site access and ligand recognition by human ABCB1" is an important contribution to the field of membrane transporter / ABC transporter structural biology. The authors determined the structures of the human ABCB1 (P-glycoprotein) in a lipid-embedded state, in apo-, substrate-bound, inhibitor-bound, and nucleotide-trapped states using cryo-EM. The structures could be determined using just the purified protein alone, in the absence of added fiducial markers / chaperones / antibodies / binders / mutations (which could potentially influence the observed protein conformations). The structures revealed obstruction of the substrate binding site by the TM4 region in the ligand free (apo) state. The authors observed dramatic conformational rearrangements in the TM helical bundle of ABCB1 induced by small molecule (substrate and inhibitor) binding, with TM4 appearing to play a crucial role in controlling the conformations of the protein.

The manuscript is very well written, and the structures are of high quality. Considering the known difficulties in working with the ABC transporters (and with ABCB1 in particular), overall the authors have done an outstanding job in analysing the data and interpreting their results. This study is particularly timely: although a number of ABC transporter (including ABCB1) structures have been resolved until now, the molecular mechanisms of substrate and inhibitor binding, as well as the conformational transitions that ABC transporter go through during ligand recognition and translocation, remain a topic of intense investigations and debates. The elegant study by Kurre et al reveals very important and previously unappreciated features of the ABCB1, and thus provides a critically important missing piece of the puzzle.

The manuscript should be acceptable for publication upon revision. I have no major concerns.

I only have minor comments, predominantly related to data presentation (figures and corresponding parts of the text that would need to be reworked) and a few minor wording issues. Addressing these points should be a relatively trivial task for the authors:

Line 199-200. The statement "Insights into TMD.. fundamentally change our understanding of how human ABCB1 works." should ideally be followed by a sentence or two clarifying exactly how our understanding changes in light of these insights. After that clarification the suggestion of new inhibitor development should follow - this would help the authors in building their arguments.

Line 207-208. ".. our data reconcile the decades of mutagenesis studies.." - it would be useful to clarify, with references, exactly which studies are reconciled by these new findings.

Methods, line 441. What was the protein/ligand molar ratio in the final sample, considering that the ligands were at 10 μ M? This would be a useful point to add to the Methods.

Methods, line 442. It is not entirely clear which of the final samples were prepared with / without ATP/Mg²⁺ added. Can this sentence be clarified?

Figure 1B. Why are the effects of taxol not shown in this figure?

Figure 1C. The structures look great, but what do the colours indicate - what do the lighter/darker shades of the colours indicate (N- and C-terminal parts of the protein)? This information should be added to the legend.

Figure 2C. The bottom right panel (with ECL3 / ECL6 labels) is very difficult to understand, because labels and arrows are all on top of each other, semi-transparent. I recommend moving the labels outside of the structure, and having arrows with some contrast - not overlaid with the structure (similar to how the labels are shown in Figure S2).

Figure 4 - TM4 and TM5 or TM4 and TM10?

Figure 5A - very busy, to the point of being confusing. It is not clear how the structures were aligned. Was only one half of the protein used as "anchor", with the other half shown as moving relative to it? Or were the structures aligned using the full range of aa residues? What effect did the authors try to convey with this comparison?

Figure 5C is more clear, but it might still be useful to visually highlight what the authors want the reader to pay attention to. In the current version it is TM4 and TM10 - perhaps this can be pointed at using an arrow / asterisk / some other label.

Figure 6. It may be useful to split this figure into 6 (panel 6A) and 7 (6B). Alternatively, 6A could be part of figure 5. In terms of the information content the current figure 6A is close to figure 5 - explaining what the current figure 5 really means.

Moving the Figure 6B panel into a separate figure will have an added advantage of increasing the figure size, allowing the reader to focus on the proposed model, and avoiding the confusion between the colours in the current 6A and 6B (red in particular).

Formatting of Figure S3 is currently suboptimal - the line numbers encroach on the figure panels. The individual panels are all of different sizes, not aligned to each other - this gives a somewhat messy impression.

Table S2 - there appears to be 6 samples on top of the table ("ABCB1 - Zosuquidar" and "ABCB1 - Taxol" - the right two columns). But then from local resolution onward the table is not filled in. Micrograph numbers are not filled in for all samples but two. Please double-check the table completeness.

Non-essential suggestions:

- Title: "insight" or "insights"?

- Figure 1A. The comparison between native mass spectrometry could be presented in parallel with size exclusion chromatography (SEC), a more traditional measure of protein quality.

- Figure 6B (especially if it eventually becomes a separate figure) could be simplified a bit by showing the labels not in the legend form, but directly in the sketches. In the current version the reader really has to familiarise themselves with the colour codes of the TMs and different elements shown in the legend, before attempting to understand what the sketches show. Making this schematic easier to read and understand would greatly improve the reader's experience.

Answers to Referees. We the reviewers' complimentary critique of our work. Both reviewers recognized the merits of our structural with reviewer 2 highlighting that our manuscript was very well written and our results of high quality, especially in consideration of the difficulties working with human ABCB1, timely, and showcase previously unappreciated features of human ABCB1 and provide 'a critical missing piece of the puzzle.' Several suggestions were made for improving the quality of our presentation and have been incorporated into our revised manuscript.

Referee#1:

From a cryo-EM point of view this is all fine. But in my opinion, we are now confronted with a dilemma. We have two fundamentally different substrate-bound structures, one in the absence and one in the presence of an inhibitory antibody. Which one is right? Without additionally biochemical evidence, no one can answer this question. Thus, the authors have to provide biochemical evidence (mutants, binding affinities, ATPase activity, etc.) that support their proposal before the manuscript can be accepted. Only then it's worthwhile to also mention minor points that deserve attention.

Answer: While the two Taxol-bound states the reviewer references have fundamentally different conformations, the majority of residues interacting with Taxol are shared between them, showcasing binding site plasticity and transformation which is known to be a hallmark feature of ABCB1. The following residues have been shown by multiple studies to have a role in substrate interactions:

1. F770((1), N721 & Q838(2), F303(1, 3, 4), F728(5, 6), and A729(7), S766(8) that were not seen to closely interact with Taxol in the Fab bound occluded state but shown to do so in the Fab-free IO structure reported here.
2. A smaller subset of residues implicated in substrate interactions like I340(9)S344(10), Q347(11) were shown to interact with Taxol in the Fab bound occluded state but not in the IO state reported here.
3. For the large number of overlapping residues between the two sites, Y307(5), Y310(12) Q725(5), Q990(11, 12) M986(5) have all been implicated in substrate interactions.
4. Finally, our IO Taxol site overlaps with the binding site for marine pollutants observed in the crystal structures of mouse ABCB1(13), lending further credence to its general conservation.

Our cryo-EM structures were determined in conditions mimicking the transporter's physiological state (mammalian brain polar lipid and cholesterol environment, no stabilizing mutants, no inhibitory Fabs that, at least for the case of UIC2, have been shown to themselves promote the Occluded state(14)) and treated identically except for the addition of ligand. The four distinct conformations observed highlight the following details that allowed us to formulate our working mechanistic model :

5. As the occluded state was not enforced by Fabs, the conformational changes observed were driven solely by the added ligand.
6. The occluded state is only induced by the inhibitor zosuquidar and not the substrate Taxol showing that while inhibitors alone can stabilize (trap) the occluded state, substrates cannot. The fact that Taxol was shown to bind to the occluded pocket in the Fab bound ABCB1

structures(3, 15) suggests that this state may be a short lived but distinct state of the substrate transport cycle.

In summary, both Taxol-bound conformational states (Ours and the fab bound state previously reported in Alam et al, Science 2019) captured the ligand in distinct states of its conformational cycle and our data actually solve the dilemma of reconciling information from the large body of published data implicating specific residues in drug interactions with structural data showing exactly how they are involved. Considering that both the residues overlapping between both sites and those distinct to each (of which there are significantly more in the IO state Taxol bound structure shown here) have already been implicated in drug binding, a redundant mutagenesis study will add nothing to the mechanistic insights provided. While a detailed analysis of binding kinetics for various drugs is indeed a long-term goal of ours and of great interest to the field, the establishment of the relevant binding assays remains well beyond the scope of this manuscript.

In our revised manuscript, we have significantly expanded the discussion of binding site residues to include the points above and added the relevant citations.

References cited and included in revision:

1. S. Vahedi, E. E. Chufan, S. V. Ambudkar, Global alteration of the drug-binding pocket of human P-glycoprotein (ABCB1) by substitution of fifteen conserved residues reveals a negative correlation between substrate size and transport efficiency. *Biochem Pharmacol* **143**, 53-64 (2017). **(Ref 43 in revision)**
2. S. Singh *et al.*, Design and synthesis of human ABCB1 (P-glycoprotein) inhibitors by peptide coupling of diverse chemical scaffolds on carboxyl and amino termini of (S)-valine-derived thiazole amino acid. *J Med Chem* **57**, 4058-4072 (2014). **(Ref 44 in revision)**
3. K. Nosol *et al.*, Cryo-EM structures reveal distinct mechanisms of inhibition of the human multidrug transporter ABCB1. *Proc Natl Acad Sci U S A* **117**, 26245-26253 (2020).
4. M. Murakami *et al.*, Second-site suppressor mutations reveal connection between the drug-binding pocket and nucleotide-binding domain 1 of human P-glycoprotein (ABCB1). *Drug Resist Updat* **71**, 101009 (2023).
5. E. E. Chufan *et al.*, Multiple transport-active binding sites are available for a single substrate on human P-glycoprotein (ABCB1). *PLoS One* **8**, e82463 (2013).
6. T. W. Loo, M. C. Bartlett, D. M. Clarke, Transmembrane segment 7 of human P-glycoprotein forms part of the drug-binding pocket. *Biochem J* **399**, 351-359 (2006).
7. H. L. Gao *et al.*, Sapitinib Reverses Anticancer Drug Resistance in Colon Cancer Cells Overexpressing the ABCB1 Transporter. *Front Oncol* **10**, 574861 (2020).
8. J. P. Becker, G. Depret, F. Van Bambeke, P. M. Tulkens, M. Prevost, Molecular models of human P-glycoprotein in two different catalytic states. *BMC Struct Biol* **9**, 3 (2009).
9. T. W. Loo, D. M. Clarke, Location of the rhodamine-binding site in the human multidrug resistance P-glycoprotein. *J Biol Chem* **277**, 44332-44338 (2002).
10. S. Iqbal *et al.*, Vinca alkaloid binding to P-glycoprotein occurs in a processive manner. *Biochim Biophys Acta Biomembr* **1864**, 184005 (2022).
11. K. Sasitharan, H. A. Iqbal, F. Bifsa, A. Olszewska, K. J. Linton, ABCB1 Does Not Require the Side-Chain Hydrogen-Bond Donors Gln(347), Gln(725), Gln(990) to Confer Cellular Resistance to the Anticancer Drug Taxol. *Int J Mol Sci* **22** (2021).

12. C. P. Wu *et al.*, Overexpression of ABCB1 and ABCG2 contributes to reduced efficacy of the PI3K/mTOR inhibitor samotolisib (LY3023414) in cancer cell lines. *Biochem Pharmacol* **180**, 114137 (2020).
13. S. C. Nicklisch *et al.*, Global marine pollutants inhibit P-glycoprotein: Environmental levels, inhibitory effects, and cocrystal structure. *Sci Adv* **2**, e1600001 (2016).
14. A. Alam *et al.*, Structure of a zosuquidar and UIC2-bound human-mouse chimeric ABCB1. *Proc Natl Acad Sci U S A* **115**, E1973-E1982 (2018).
15. A. Alam, J. Kowal, E. Broude, I. Roninson, K. P. Locher, Structural insight into substrate and inhibitor discrimination by human P-glycoprotein. *Science* **363**, 753-756 (2019).

Referee#2:

Line 199-200. The statement "Insights into TMD.. fundamentally change our understanding of how human ABCB1 works." should ideally be followed by a sentence or two clarifying exactly how our understanding changes in light of these insights. After that clarification the suggestion of new inhibitor development should follow - this would help the authors in building their arguments.

Answer: We have reworked the discussion as suggested for greater and in light of the expanded discussion of Taxol interacting residues mentioned above. (Lines 162-214 of revision)

Line 207-208. ".. our data reconcile the decades of mutagenesis studies.." - it would be useful to clarify, with references, exactly which studies are reconciled by these new findings.

Related to the point above, we have included details of the residues previously implicated in substrate interactions and significantly expanded the discussion of binding site residues as described above in response to reviewer 1.

(Methods, line 441. What was the protein/ligand molar ratio in the final sample, considering that the ligands were at 10 uM? This would be a useful point to add to the Methods.

Answer: We have now clarified this point and added the following missing information (line 450-454 in revision):

Pooled peak fractions from SEC at a concentration of ~0.15 mg/ml (1uM) were mixed with a 10x molar excess of Taxol or zosuquidar with or without ATP/Mg²⁺ and incubated together before concentrating the samples to 0.5-1mg/ml (~ 3-6 uM) for grid preparation similar to prior studies (21).

Methods, line 442. It is not entirely clear which of the final samples were prepared with / without ATP/Mg²⁺ added. Can this sentence be clarified?

Answer: We have modified the relevant methods section for clarity. In short, both the final Taxol bound and zosuquidar bound states were determined in the presence of ATP/Mg²⁺. In addition, we have also submitted the maps of the same complexes without added nucleotides but these do not have associated pdb coordinates and are not discussed in detail in the manuscript, only highlighting the overall similarity in observed conformations.

1. ABCB1 + Taxol + ATP/Mg²⁺
2. ABCB1 + Zosuquidar + ATP/Mg²⁺
3. ABCB1 + ATP γ S/Mg²⁺
4. ABCB1 + Taxol
5. ABCB1 + Zosuquidar

Figure 1B. Why are the effects of Taxol not shown in this figure?

Answer: We have now included additional data showing the effect of Taxol on ATPase in our new Figure 1B.

“**Figure 1 Conformational landscape of lipid embedded human ABCB1. A** Comparison of saposin and nanodisc reconstituted human ABCB1 by nMS and the normalized HPLC chromatograms of both are showing in the top right corner. **B** Comparison of ATPase activity of saposinA, MSP1D1 nanodisc, and Liposome reconstituted human ABCB1 in presence of inhibitor, Zosuquidar (solid shapes and lines) and Taxol (clear shapes and dashed lines), basal ATPase rates are shown in black dashed box. N=3 and error bars denote standard deviation. **C** Structures of human ABCB1 in multiple distinct conformational states. EM density for the two halves is colored differently with N-terminal half (half1) in lighter shade and C-terminal half (half2) in darker shade and that of modeled acyl chains is colored gray.”

Figure 1C. The structures look great, but what do the colours indicate - what do the

lighter/darker shades of the colours indicate (N- and C-terminal parts of the protein)? This information should be added to the legend.

Answer: That is correct, and we have now updated our figure legend accordingly.

Figure 2C. The bottom right panel (with ECL3 / ECL6 labels) is very difficult to understand, because labels and arrows are all on top of each other, semi-transparent. I recommend moving the labels outside of the structure, and having arrows with some contrast - not overlaid with the structure (similar to how the labels are shown in Figure S2).

Answer: Figure 2 has now been updated as suggested.

“Figure 2. Structure of apo-ABCB1. A Overall structure with the two halves colored as different shades of red and density modeled as lipid acyl chains (gray sticks) shown as transparent gray surfaces. **B** 3TM bundle formation by TM4, TM6, and TM12. TM4 sub-helical segments. The yellow dashed triangle highlights the central 3TM bundle in top and bottom views. **C** Comparison of the cryo-EM structure of apo-ABCB1, colored as in A, and its alphaFold predicted structure (transparent cartoon). Black arrows indicate major movements of select TMs. The gray bars represent the plasma membrane.”

Figure 4 - TM4 and TM5 or TM4 and TM10?

Answer: We thank the reviewer for pointing out this typo, and have corrected Figure 4B to correctly show TM10.

Figure 5A - very busy, to the point of being confusing. It is not clear how the structures were aligned. Was only one half of the protein used as "anchor", with the other half shown as moving relative to it? Or were the structures aligned using the full range of aa residues? What effect did the authors try to convey with this comparison?

Answer: We have significantly modified figure 5 for greater clarity and, as suggested by the reviewer, replaced panel 5A with panel 6A from our original submission. The overall structural alignments between each conformational state are now shown as Figure EV3 with information of number r.m.s.d information included. These alignments were done using the full range of residues.

“Figure 5 Structural Transitions in ABCB1 focused non the . A Overlay of TM4/5 and TM10/11 of all structures reported, highlighting overall conformational changes linked to substrate (Taxol molecule, pink surface map) or inhibitor (zosuquidar molecule, teal surface map) binding and CH2 and CH4 movements (bottom) with distances between selected Ca pairs shown. **B** Pairwise structural alignment of linked TM pairs expected to move together in different type II ABC exporter conformational states with TM4/5 and TM10/11 pairs boxed to highlight their greater conformational flexibility in the four conformations reported.

Figure EV3

“Figure EV3 Overlay of different conformational state of ABCB1. Overall structural alignments between each conformation. R.m.s.d values are also shown for total and aligned C alpha pairs

Figure 5C is more clear, but it might still be useful to visually highlight what the authors want the reader to pay attention to. In the current version it is TM4 and TM10 - perhaps this can be pointed at using an arrow / asterisk / some other label.

Answer: Figure 5C (new Figure 5B) has been modified to highlight the greater range of motion of TM4/5 and TM10/11 pairs (boxed) observed in the four conformations of human ABCB1 reported here.

Figure 6. It may be useful to split this figure into 6 (panel 6A) and 7 (6B). Alternatively, 6A could be part of figure 5. In terms of the information content the current figure 6A is close to figure 5 - explaining what the current figure 5 really means.

Answer: Figure 6 now only includes the mechanism panel (old panel 6B) that has been updated for greater clarity as suggested by the reviewer.

“Figure 6 Mechanism of ABCB1 transport function. Schematic of working model for substrate transport and inhibition in human ABCB1. With the exception of the OF_{OPEN} state (based on homologous transporters like human ABCD1(16) and Sav1866(17)). Green circles highlight potential intermediate/alternate states.”

Moving the Figure 6B panel into a separate figure will have an added advantage of increasing the figure size, allowing the reader to focus on the proposed model, and avoiding the confusion between the colours in the current 6A and 6B (red in particular).

Answer: This has been addressed above with the mechanism figure now separated (new Figure 6 in revision).

Formatting of Figure S3 is currently suboptimal - the line numbers encroach on the figure panels. The individual panels are all of different sizes, not aligned to each other - this gives a somewhat messy impression.

Answer: We apologize for these errors. In Figure EV4, text and figures have now been arranged for better clarity.

Table S2 - there appears to be 6 samples on top of the table ("ABCB1 - Zosuquidar" and "ABCB1 - Taxol" - the right two columns). But then from local resolution onward the table is

not filled in. Micrograph numbers are not filled in for all samples but two. Please double-check the table completeness.

Answer: Micrograph numbers and other details of ABCB1_{zosuquidar} and ABCB1_{Taxol} are now filled in Expanded view table 2.

Non-essential suggestions:

- Title: "insight" or "insights"?

Answer: As suggested, "insight" has been changed to "insights" in the title.

- Figure 1A. The comparison between native mass spectrometry could be presented in parallel with size exclusion chromatography (SEC), a more traditional measure of protein quality.

Answer: HPLC chromatograms of ABCB1 in saposinA and MSP1D1 nanoparticles are now added in Figure 1A at the top right corner as shown below.

“

- Figure 6B (especially if it eventually becomes a separate figure) could be simplified a bit by showing the labels not in the legend form, but directly in the sketches. In the current version

the reader really has to familiarise themselves with the colour codes of the TMs and different elements shown in the legend, before attempting to understand what the sketches show. Making this schematic easier to read and understand would greatly improve the reader's experience.

Answer: Figure 6 has now been completely revised to only include the mechanistic scheme, which has been modified for greater clarity as suggested by the reviewer.

“Figure 6 Mechanism of ABCB1 transport function. Schematic of working model for substrate transport and inhibition in human ABCB1. With the exception of the OF_OPEN state (based on homologous transporters like human ABCD1(16) and Sav1866(17)). Green circles highlight potential intermediate/alternate states.”

Dear Amer,

Thank you for submitting the revised version of your manuscript, which addresses the concerns of the referees. This revised version has now been re-reviewed; I attach the second referee reports to the bottom of this mail. As you will see, you have addressed the referees' concerns to their satisfaction. Reviewer 2 makes some final constructive suggestions which I would like you to consider carefully. Before I can finally accept the manuscript, there are some remaining editorial points which need to be addressed. In this regard, would you please:

- acknowledge funding from the Hormel Foundation; the Eagles Telethon postdoctoral fellowship; University of Minnesota start-up funds in our online submission system,
- include up to five keywords,
- list references in alphabetical order, compile references into one section and place before figure legends, for long author lists, use 10 authors + et al.,
- rename the Conflict of Interest section the "Disclosure Statement and Competing Interests" statement,
- remove the author credit section from the manuscript file,
- rename callouts as Table EV1-EV2, instead of EV table 1-2 and include callouts for Fig. EV2A-B,
- include only section names should be in the third column (the pink one) of the author checklist (not explanations),
- complete the uploaded blank SD checklist,
- ensure that reviewer access codes for EMDB datasets (EMD-45854, EMD-45904, EMD-45903, EMD-45906, EMD-45931, EMD-45932) and PDB (9CR8, 9CTF, 9CTC, 9CTG) are publicly available and provided in the data availability statement,
- describe the nature of entity for 'n' in the legends of figure 1B,
- define the centre of error bars in the legend of figure 1B
- limit the number of EV figures to five,
- upload EV tables as individual files, and remove EV table legends from the manuscript file,
- remove Additional information, Supplementary information section and EV table legends from the manuscript file, and
- correct the section order as follows: Section order should be corrected: Title page - Abstract & Keywords - Introduction - Results - Discussion - Methods - Data Availability - Acknowledgements - Disclosure and Competing Interests Statement - References - Figure Legends - Table(s) - Expanded View Figure Legends.

I look forward to receiving these changes. EMBO Press is an editorially independent publishing platform for the development of EMBO scientific publications.

Best wishes,

William

William Teale, PhD
Editor
The EMBO Journal
w.teale@embojournal.org

We realize that it is difficult to revise to a specific deadline. In the interest of protecting the conceptual advance provided by the work, we recommend a revision within 3 months (3rd Mar 2025). Please discuss the revision progress ahead of this time with the editor if you require more time to complete the revisions. Use the link below to submit your revision:

Referee #1:

Reading the revised version of the manuscript, especially the discussion section, made it clear to me that there are residues present in both binding modes (with and without Fab), only in the Fab, and only in the mode described in this manuscript. Since the number of residues interacting in both binding modes is the largest number, I can understand that mutagenesis studies will likely not resolve the dilemma. Nevertheless, I am surprised that the authors did not even try to address my point. Although, I also have to admit that the chances of success in light of the large number of mutations that have to be introduced at the same time are very small. Therefore, I am satisfied with the revised version and recommend acceptance of the manuscript.

Referee #2:

The revised manuscript is substantially improved. I think the authors did a great job, addressing all criticisms in a satisfactory manner.

I only have only one comment to the content. It is a minor point, and I think it is non-essential, but the authors might find it useful: Figure 6 looks very good now. But I think the legend is minimal to the point of being useless - I am sure the authors will agree that this legend is just too short. I think the authors could make Figure 6 greater still by adding a brief but instructive description in the legend. A few sentences describing briefly each state and how they are connected, as shown graphically in the figure, would be a powerful addition to this otherwise excellently crafted illustration.

And one more minor point:

There is at least one instance on page 7 of the response that looks like some sort of a copy-paste error, as the main text shows something else: "Figure 5 Structural Transitions in ABCB1 focused non the.". I would advise the authors to carefully go through all figures, legends and text, to be sure there are no similar copy-paste errors. If this happened in the response letter, it may have happened elsewhere.

All editorial and formatting issues were resolved by the authors.

Dear Amer,

I am pleased to inform you that your manuscript has been accepted for publication in the EMBO Journal.

Congratulations to you and your team on these beautiful structures!

Best wishes,

William

William Teale, PhD
Editor
The EMBO Journal
w.teale@embojournal.org
